# UNDERSTANDING THE FUNCTIONAL AND STRUCTURAL DIFFERENCES ACROSS EXCITATORY AND INHIBITORY NEURONS

## ABSTRACT

One of the most fundamental organizational principles of the brain is the separation of excitatory (E) and inhibitory (I) neurons. In addition to their opposing effects on post-synaptic neurons, E and I cells tend to differ in their selectivity and connectivity. Although many such differences have been characterized experimentally, it is not clear why they exist in the first place. We studied this question in deep networks equipped with E and I cells. We found that salient distinctions between E and I neurons emerge across various deep convolutional recurrent networks trained to perform standard object classification tasks, and explored the necessary conditions for the networks to develop distinct selectivity and connectivity across cell types. We found that neurons that project to higher-order areas will have greater stimulus selectivity, regardless of whether they are excitatory or not. Sparser connectivity is required for higher selectivity, but only when the recurrent connections are excitatory. These findings demonstrate that the functional and structural differences observed across E and I neurons are not independent, and can be explained using a smaller number of factors.

Deep neural networks have become powerful tools to model the brain (Yamins & DiCarlo, 2016; Kriegeskorte, 2015). They have been used to successfully model various aspects of the sensory (Yamins et al., 2014; Kell et al., 2018), cognitive (Mante et al., 2013; Wang et al., 2018; Yang et al., 2019), and motor system (Sussillo et al., 2015). Deep networks have been particularly effective at capturing neural representations in higher-order visual areas that remain challenging to model with other methods (Yamins et al., 2014; Khaligh-Razavi & Kriegeskorte, 2014).

Demonstrating that these models can mimic the brain in some respects is only a starting point in the process of advancing neuroscience with deep learning. Once some relationship between artificial and biological networks is established, the model can be dissected to better understand *how* certain biological computations are mechanically implemented (Sussillo & Barak, 2013). Although notoriously difficult to interpret, deep networks are still much more accessible than the brain itself. This is how deep networks can help address the "how" question. On top of that, deep networks can be used in a normative approach to answer the "why" question. Suppose we found that a certain architecture, objective function (dataset), and training algorithm could together lead to a neural network matching some features of the brain. Then we can ask why the brain evolved such features by testing which element of the architecture, dataset, and training is essential for such features to evolve in the artificial networks. For example, Lindsey et al. (2019) showed early layers of a convolutional neural network can recapitulate the center-surround receptive field observed in retina, but only when followed by an information bottleneck similar to the one that exists from retina to cortex. This finding demonstrates how realistic biological constraints can be used to explain the emergence of known properties of the brain.

Despite the many successes of applying deep networks to model the brain, standard architectures deviate from the brain in many important ways. Recently, neural networks that incorporate fundamental structures of the brain are becoming increasingly common (Kar et al., 2019; Miconi et al., 2018; Song et al., 2016). One fundamental structural principle of the brain is the abundance of recurrent connections within any cortical area. Cortical neurons receive a substantial proportion of their inputs from other neurons in the same area (Harris & Shepherd, 2015). Convolutional recurrent networks trained on object classification tasks can perform comparably to purely feedforward

networks with similar numbers of parameters, while being able to better explain temporal responses in higher visual areas (Nayebi et al., 2018; Kar et al., 2019).

One of the most fundamental organizational principles of the cortex is the separation of excitatory (E) and inhibitory (I) neurons (Dale, 1935; Eccles et al., 1954). Dale's law states that each neuron expresses a fixed set of neurotransmitters, which results in either excitatory or inhibitory downstream effects. In addition to the difference in their immediate impacts on post-synaptic neurons, E and I neurons differ in several other important ways. There are several times (4-10x) more E neurons than I neurons (Hendry et al., 1987). Neurons that project to other areas, the so-called "principal neurons", are all excitatory in the cortex (Bear et al., 2007). In the mammalian sensory cortex, E neurons are overall more selective to stimuli than I neurons in the same area as extensively reported in mice (Kerlin et al., 2010; Znamenskiy et al., 2018) and to a lesser extent in other species (Wilson et al., 2017). Finally, E neurons are more sparsely connected with each other, with a connection probability of around 10% (Harris & Shepherd, 2015), compared to I neurons, which target almost all neighboring E neurons (Pfeffer et al., 2013).

Although many of these differences across E and I neurons are well-known and routinely incorporated in computational models (Somers et al., 1995; McLaughlin et al., 2000), it remains unclear whether they can emerge in deep neural networks from training. It is also unknown whether these differences are all independent properties, each contributing to the computation, or if some differences are natural results of the others. To answer these questions, we first trained many variants of deep convolutional recurrent neural networks on classical object classification tasks. Across all variants, three structural principles are built in: Dale's law, the abundance of excitatory neurons, and the role of excitatory units as principal neurons. However, other observed properties of biological E-I circuits are not hardwired, and therefore could only evolve under the pressure of performing the task. We found that functional and structural differences across E and I neurons emerged through training. The development of these characteristics allows us to address the "why" question by removing the built-in differences across E and I neurons and monitoring whether specific functional and structural properties still emerge. Our anonymized code is available at https://anonymous.4open.science/repository/d0ae905f-4171-42b0-94b5-abf03d6414aa.

## 1 MULTI-CELL CONVOLUTIONAL RECURRENT NETWORKS

The networks we use to model the visual cortex consist of 2 layers of purely feedforward, convolutional processing, followed by 2 or 3 layers of recurrent processing (Figure 1). The feedforward layers correspond loosely to retina and thalamus, while the recurrent layers correspond to cortex. Each recurrent layer consists of excitatory and inhibitory neurons (channels).

We examined many variants for the recurrent layer. For brevity, here we focus on two representative architectures. The first one, which for convenience we refer to as the *StandardEI* model, consists of recurrently connected E and I channels:

$$
\begin{aligned}
\text{Curr}_E &= W_{X \to E} * X_t + W_{E \to E} * E_{t-1} - W_{I \to E} * I_{t-1} + b_E \\
\text{Curr}_I &= W_{X \to I} * X_t + W_{E \to I} * E_{t-1} - W_{I \to I} * I_{t-1} + b_I \\
E_t &= f_E \circ E_{t-1} + (1 - f_E) \circ \sigma_c(\text{Curr}_E) \\
I_t &= f_I \circ I_{t-1} + (1 - f_I) \circ \sigma_c(\text{Curr}_I)
\end{aligned}
\tag{1}
$$

$X_t$ is the input from the previous layer, $E_t$ and $I_t$ are the activity of the excitatory and inhibitory neurons. $\text{Curr}_E$ and $\text{Curr}_I$ are the input currents for E and I neurons respectively. Batch normalization is applied to $\text{Curr}_E$, and we tested variants where batch normalization is also applied to $\text{Curr}_I$. $f_E$ and $f_I$ are forget gates, implemented as $\sigma(\widetilde{f})$ where $\sigma$ is the sigmoid function and $\widetilde{f}$ is a trainable 1-d tensor with the same size as the number of channels. Dale's law is implemented by setting all relevant weight matrices $W_{E \to \cdot}, W_{I \to \cdot}$ to be non-negative using an absolute function, $W = |\widetilde{W}|$, where the trainable variable is the non-sign-constrained weight matrix $\widetilde{W}$. $b_E$ and $b_I$ are bias terms. We found that task performance drops substantially if a rectified linear function is used to impose the sign constraint instead. Only excitatory neurons in each recurrent layer makes long-range connections to the next layer, making them the principal neurons (PN). In comparison, inhibitory neurons are all interneurons (Int), making within-layer connections.

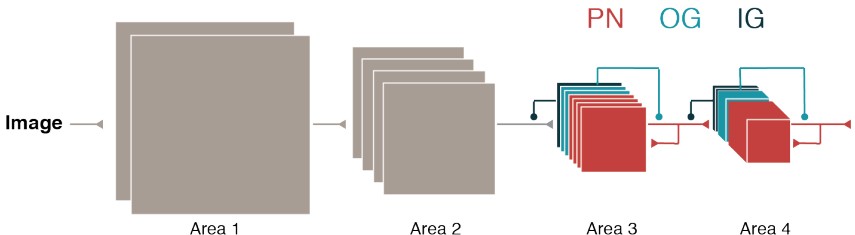

**Figure 1: Convolutional recurrent network with multiple cell types.** The *Standard* model for CIFAR10 starts with two layers of regular convolutional processing, followed by 2 layers of recurrent processing. Each recurrent layer consists of three types of cells. The excitatory principal neurons (PN) output non-negative connection weights and target the next area. Output-gating (OG) and input-gating (IG) neurons are inhibitory and only target neurons within the same area.

In the brain, E and I neurons can be further divided into many subtypes that differ in their inputs, output targets, and gene expression (Markram et al., 2004). Two major types of inhibitory neurons account for 60% of all inhibitory neurons in the cortex (Rudy et al., 2011). The first type expresses the molecule Somatostatin (SST) and inhibits dendrites of E neurons, the structure that receives inputs from other neurons. The second type expresses the protein Parvalbumin (PV) and inhibits soma of E neurons, where outputs are generated. Experimental evidence suggested that these input- and output-controlling inhibition can function like subtractive or multiplicative gates (Lee et al., 2012; Wilson et al., 2012). As was previously pointed out (Costa et al., 2017), this motif in which principal neurons–those that can project to other areas–are recurrently controlled by multiple gates is reminiscent of the design of common recurrent units in machine learning, including Long Short-Term Memory (LSTM) (Hochreiter & Schmidhuber, 1997) and Gated Recurrent Unit (GRU) (Cho et al., 2014) networks.

Motivated by these findings, we introduced a convolutional recurrent layer with two distinct types of inhibitory interneurons (Int), which we refer to as the *Standard* model. These two types of neurons control/gate the input and output of the excitatory neurons respectively, and are referred to as the input-gating (IG) and output-gating (OG) neurons. Taken together, *Standard* model is described by

$$
\begin{aligned}
f_t &= \sigma_f(W_{X \to f} * X_t + W_{\text{PN} \to f} * \text{PN}_{t-1} + b_f), \\
\text{IG}_t &= \sigma_c(W_{X \to \text{IG}} * X_t + W_{\text{PN} \to \text{IG}} * \text{PN}_{t-1} + b_{\text{IG}}), \\
i_t &= \sigma_i(W_{\text{IG} \to \text{PN}} * \text{IG}_t + b_i), \\
\text{OG}_t &= \sigma_c(W_{X \to \text{OG}} * X_t + W_{\text{PN} \to \text{OG}} * \text{PN}_{t-1} + b_{\text{OG}}), \\
o_t &= \sigma_o(W_{\text{OG} \to \text{PN}} * \text{OG}_t + b_o), \\
C_t &= f_t \circ C_{t-1} + (1 - i_t) \circ \sigma_c(W_{X \to \text{PN}} * X_t + W_{\text{PN} \to \text{PN}} * \text{PN}_{t-1} + b_c), \\
\text{PN}_t &= \sigma_c(C_t - o_t).
\end{aligned}
\tag{2}
$$

Here $\text{IG}_t, \text{OG}_t, \text{PN}_t$ are the activity of the input-gating, output-gating, and principal neurons. $C_t$ is the pre-activation values (or membrane potential) of the principal neurons, corresponding to the cell state in LSTM. Batch normalization is applied to $C_t$. IG and OG neurons inhibit PNs through inhibitory currents $i_t$ and $o_t$. $f_t$ stands for the forget gate, similar to LSTM. All neurons are rectified linear units (ReLU) (see Appendix B for justification), $\sigma_c = \max(x, 0)$. The nonlinearity of the gate variable is the sigmoid function for multiplicative and forget gates ($\sigma_i, \sigma_f$), and ReLU for subtractive gates ($\sigma_o$). $*$ stands for convolution, while $\circ$ stands for element-wise multiplication. The *Standard* model is closely related to LSTM in its use of forget, input, and output gates. However, in LSTM, there is only a single type of neurons, the principal neurons, from which the input/output gates are directly generated.

## 2 REPRODUCING FUNCTIONAL AND STRUCTURAL DIFFERENCES ACROSS CELL TYPES

In this section, we will show that several experimentally-observed functional and structural differences across excitatory (PN) and inhibitory (Int) neurons can naturally emerge in E-I networks

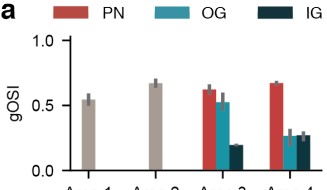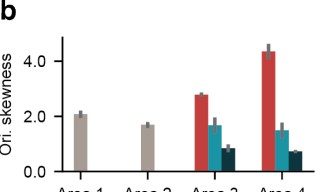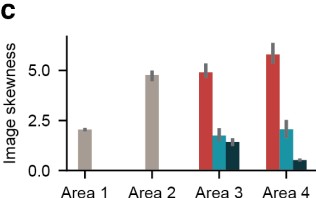

**Figure 2: Model excitatory neurons have higher orientation selectivity and natural image selectivity than inhibitory neurons.** (a,b) The average gOSI (a), orientation skewness (b), and image skewness (c) for each type of neurons in the *Standard* networks. Error bar is the 95% confidence interval computed with bootstrapping across 5 networks.

trained on image classification tasks. These results will be illustrated mainly in the *Standard* model, however, they hold for all variants of E-I networks tested, including the *StandardEI* model.

We trained the *Standard* model on the image classification datasets CIFAR10 (Krizhevsky & Hinton, 2009) and ImageNet (Deng et al., 2009). We used common training techniques including momentum (Polyak, 1964) and L2 regularization. For CIFAR10, regularization coefficient is 0.0002, the initial learning rate is 0.1 and decays 10-fold at epochs 100, 150, and 200, in all 250 epochs. The network consists of two convolutional feedforward layers of 16, and 32 channels each, followed by two recurrent layers. The first recurrent layer contains 64 PN, 16 IG, and 16 OG channels. For each type of neurons, the number of channels is doubled in the second recurrent layer. The network is unfolded for 4 time steps, and the classification output is read-out from the final time step using a fully connected linear layer. For all conditions, we trained 5 networks with different random seeds. The network reaches approximately 85% test accuracy on CIFAR10, comparable to convolutional networks of similar depth (Hinton et al., 2012). See Appendix A for more details and hyperparameters for ImageNet.

We ask whether training leads to qualitative differences between excitatory and inhibitory neurons in the network, as observed in the brain. In mouse cortex, inhibitory neurons are functionally less selective than excitatory neurons (Kerlin et al., 2010; Znamenskiy et al., 2018), meaning that they tend to respond to different stimuli with similar values. We first measured the selectivity to static oriented gratings for excitatory and inhibitory neurons (Figure 8a). The selectivity is quantified in two ways. First, we computed the global Orientation Selectivity Index (gOSI) for each neuron $j$ (Wörgötter & Eysel, 1987),

$$\text{gOSI}_j = \Big| \sum_{k=1}^{M} r_j(\theta_k) e^{i2\theta_k} / \sum_{k=1}^{M} r_j(\theta_k) \Big|, \tag{3}$$

where $r_j(\theta_k)$ is the $j$-th neuron's response to the $k$-th grating stimulus at orientation $\theta_k$. gOSI is close to 1 when the neuron is highly selective to orientation. For each type of neurons, we report the average gOSIs across all channels in the center of the convolutional layer. The second measure of selectivity is the skewness of the distribution of responses to various stimuli (Samonds et al., 2014). The skewness $\gamma_j$ for neuron $j$ is defined as

$$\gamma_j = \frac{\frac{1}{M} \sum_{k=1}^{M} (r_{j,k} - \bar{r}_j)^3}{[\frac{1}{M} \sum_{k=1}^{M} (r_{j,k} - \bar{r}_j)^2]^{3/2}}, \tag{4}$$

where $r_{j,k}$ is the $j$-th neuron's response to the $k$-th stimulus, and $\bar{r}_j$ is the average response. The skewness is higher if a neuron is strongly selective to a small number of stimuli. For both gOSI (Figure 2a) and orientation skewness (Figure 2b), the selectivity is substantially higher for excitatory neurons (PN) compared to inhibitory neurons (IG, OG), in both recurrent layers (areas 3 and 4). Next, we measured the selectivity to natural images. We computed the skewness of responses to all images in the test set of CIFAR10. Again, excitatory neurons in both areas 3 and 4 have higher selectivity to natural images than inhibitory neurons (Figure 2c).

Another major qualitative difference between excitatory and inhibitory neurons in the brain is the higher density of I-to-E connections (i.e., Int-to-PN) compared to E-to-E (i.e., PN-to-PN). In our

network, we found that the distribution of all recurrent connection weights is multi-modal (Figure 3a), with a group of strong connections and many much weaker connections. To assess the connection density, we chose a threshold at $\exp(-10)$, and quantified the proportion of connection weights that exceed this threshold. This threshold is chosen to separate the strongest mode from the weaker modes in the distribution of all weights (Figure 3a). The distribution of PN-to-PN connections is spread out (Figure 3b), leaving a substantial proportion of connections below the threshold. In contrast, the distributions of both IG-PN and OG-PN connection weights are mostly concentrated above the threshold (Figure 3c,d), leading to higher connection density (Figure 3e). Therefore, sparser excitatory connectivity emerged in the network after training, in agreement with biological observations.

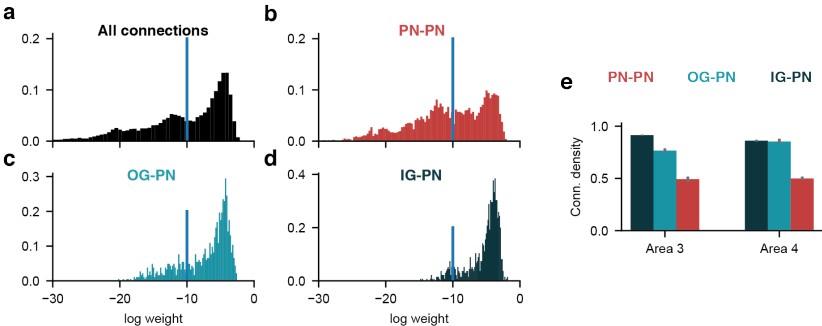

**Figure 3: Model excitatory neurons are more sparsely connected.** (a-d) The distribution of connection weights for all recurrent connections (a), and for the PN-to-PN (b), OG-to-PN (c), and IG-PN connections (d) in area 3 of one standard network. Blue line: the threshold used to compute the proportion of strong connections, namely those that exceed the threshold. (e) The connection density (proportion of strong connections) for three types of excitatory and inhibitory connections across area 3 and 4.

The findings above are summarized in Figure 4, which also includes results from the *StandardEI* model, and from networks trained on ImageNet. The emergent differences across E and I neurons are in qualitative agreement with experimental estimates. To test the generality of these results, we trained networks with subtractive or multiplicative gates (Figure 9a), networks that structurally interpolate between the *Standard* and *StandardEI* models (Figure 9b), and networks with or without batch normalization or dropout (Figure 9c,17). In all variations of networks that reached accuracy above 80%, the excitatory neurons are more selective and less connected.

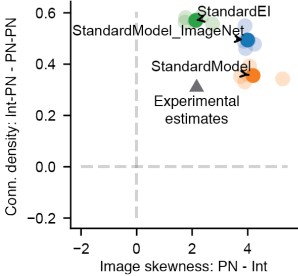

**Figure 4: The E-I differences emerge across network variations and datasets.** The E-I differences are summarized using the difference in image skewness between principal neurons (PN) and interneurons (Int), and the difference in connection density between Int-to-PN and PN-to-PN connections. Light circles: individual networks. Dark circles: average. Experimental estimates are derived from Znamenskiy et al. (2018), Song et al. (2005), and Packer & Yuste (2011) (Appendix E).

By monitoring the orientation selectivity, image selectivity, and connection density throughout the training process, we found that the differences across excitatory and inhibitory cells emerged early on (Figure 5). Even though excitatory and inhibitory neurons started out with similar selectivity

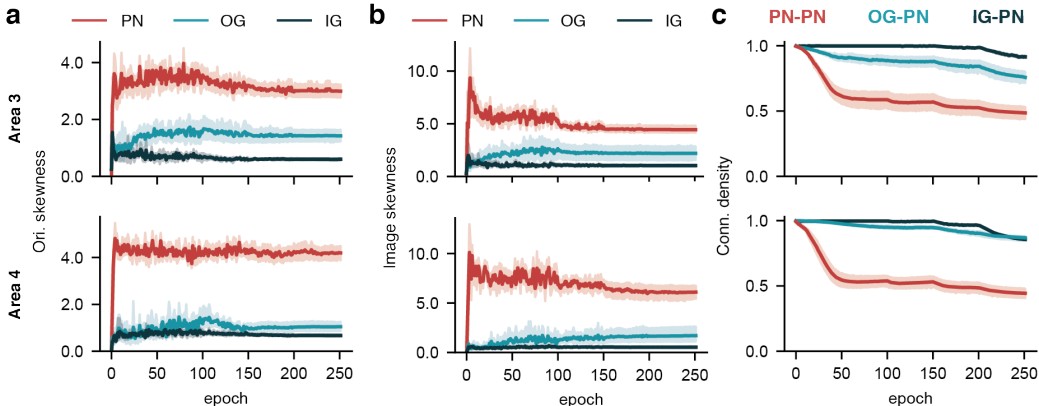

**Figure 5: The E-I differences emerge early in training.** The orientation skewness (a), image skewness (b), and connection density (c) for areas 3 (top) and 4 (bottom) throughout training.

(close to 0), and similar connectivity (close to 1), the differences across cell types become substantial after 5-30 epochs. These functional and structural differences remain stable throughout training, even as the training performance continues to improve (Figure 10). Together, these results argue that there exists a strong optimization pressure to differentiate the selectivity and connectivity of excitatory and inhibitory neurons.

# 3 WHY DO EXCITATORY AND INHIBITORY NEURONS HAVE DISTINCT SELECTIVITY AND CONNECTIVITY?

We have shown that neural networks with excitatory and inhibitory neurons develop different selectivity and connectivity, qualitatively reproducing long-standing findings in the brain. Now we ask why these differences emerge.

The built-in structural asymmetry between our model E and I neurons must be critical for their emergent differences. We have incorporated three major forms of asymmetry that exist in the brain. First, there is an asymmetry in numbers. Excitatory neurons are 4 to 10 times more abundant than inhibitory neurons in the brain. In all models tested so far, there are more excitatory than inhibitory neurons in each area. Second, there is an asymmetry in projection targets. In the cortex, all principal neurons are excitatory. Meanwhile, all inhibitory neurons are interneurons, meaning that they only connect to other neurons within the same area. Third, there is by definition an asymmetry in action. Excitatory neurons excite other neurons while inhibitory neurons inhibit. When the activation function of a neuron is rectified and non-saturating (for example, ReLU), excitatory inputs to the neuron can move it into a responsive regime, where inhibitory inputs can make a neuron non-responsive. In this section, we will remove individual asymmetry, and test which one led to the observed differences in selectivity and connectivity. In the *Standard* model, there is an additional asymmetry because inhibitory neurons use multiplicative gates to control inputs and outputs of excitatory neurons. This asymmetry is not presented in the *StandardEI* model. Therefore, we will present results based on variations of both the *Standard* and the *StandardEI* model trained on CIFAR10.

**Asymmetry in numbers** In our *Standard* model, the ratio between the number of OGs(IGs) and PNs is 1:4. It is conceivable that excitatory neurons can afford to be more selective to orientations and images because there are more of them. To test this hypothesis, we independently varied the numbers of E and I channels in the recurrent layers. The number of E or I channels ranges from 2 to 256 in area 3. Area 4 always has the doubled number of channels. We found that the number of E, but not I, channels have a strong influence on both image selectivity (Figure 6a, left) and connection density (Figure 6b, left) of E neurons. In contrast, the inhibitory selectivity is consistently low, regardless of the number of E or I channels (Figure 6a, right). The inhibitory connection density depends moderately on the number of E and I channels, decreasing with a larger number of I channels (Figure 6b, right).

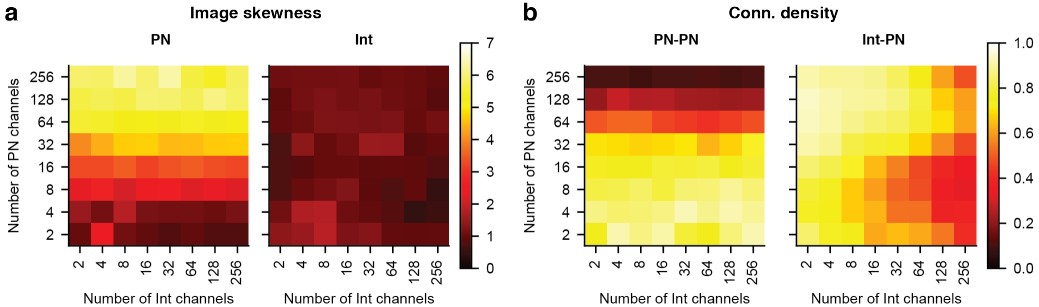

**Figure 6: Selectivity and connectivity in networks with various numbers of excitatory and inhibitory channels.** (a) The image skewness of principal neurons (PN, left) and interneurons (Int, right) for networks of various numbers of excitatory and inhibitory channels. (b) The connection density of PN-PN connections and Int-PN connections for various networks. Results from areas 3 and 4, and from two types of interneurons are combined. All networks shown have accuracy above 60%.

Even though excitatory neurons become more selective and less connected when there are more of them, the asymmetry in numbers alone does not explain the emergent E-I differences. When the number of excitatory and inhibitory channels are the same and both high, excitatory neurons still have higher selectivity and sparser connectivity. Similar to selectivity and connectivity, task accuracy depends mainly on the number of excitatory channels (Figure 11), meaning that it is not necessary to have a high number of inhibitory channels in these networks. Without any inhibitory channels, task accuracy only drops moderately in our *Standard* and *StandardEI* models. However, without batch normalization, accuracy drops to chance level when recurrent layers consist of only excitatory neurons (see Figure 17 and Appendix L). This potentially explains why there are 4-8 times less inhibitory neurons than excitatory neurons in the brain.

**Asymmetry in projection**    In both the cortex and all networks tested so far, principal neurons are exclusively excitatory. Here, we decoupled this relationship by training *InhPN* networks, where long-range connections originate from inhibitory, instead of excitatory, neurons. We do so by flipping the sign of all local recurrent connections. The connections from principal neurons to the next layer are kept excitatory, because we had difficulty training networks with inhibitory long-range projections past 80% accuracy. *InhPN* networks can still achieve accuracy similar to that of the standard network. However, the now-inhibitory principal neurons (PN) remain more selective to orientation and natural images compared to the now-excitatory interneurons (Int) (Figure 7, green). This result argues that higher selectivity is not a property inherent to excitatory neurons. Whichever type of neurons serves as the principal neurons would demand higher selectivity, presumably because the principal neurons need to carry detailed stimulus information to the next layer.

Interestingly, in the *InhPN* networks, the connectivity among inhibitory principal neurons (PN) become dense despite maintaining high selectivity (Figure 7, green). This result is in stark contrast with the sparser connectivity needed for highly selective excitatory neurons in the *Standard* model (Figure 6). We speculate that these seemingly contradictory results could potentially be understood in the context of recurrently connected ReLU neurons. When a ReLU neuron is highly selective, typically, it is strongly activated by only a small subset of stimuli. If this selectivity is supported by recurrent excitatory connections, then a neuron only needs to receive inputs from a small set of excitatory neurons with similar preferred stimuli. Meanwhile, if the selectivity is supported by recurrent inhibition, then a neuron needs to receive inputs from nearly all other neurons, except for the small subset of neurons with similar preferences. Therefore, recurrent connections between highly selective inhibitory neurons should be dense, instead of sparse.

**Asymmetry in action**    The namesake difference between excitatory and inhibitory neurons is the sign of their connection weight values. All connections stemming from an excitatory (inhibitory) neuron are constrained to be non-negative (non-positive). We can release this constraint on the sign of connection weights, while keeping other E-I asymmetries. Such *NoConstraint* networks

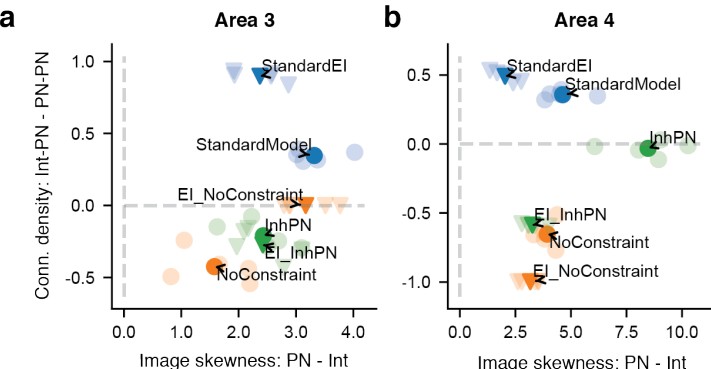

**Figure 7: Removing asymmetry between excitatory and inhibitory neurons.** Difference in connection density against difference in image selectivity for areas 3 (a) and 4 (b) for the original networks (blue), their *InhPN* variants (green), and *NoConstraint* variants (orange). Light symbols: individual networks; dark symbols: average. Circle: *Standard* model and its variants; triangle: *StandardEI* and its variants.

achieve slightly better accuracy compared to the Dale's law-obeying networks. In the *NoConstraint* networks, the formerly-excitatory principal neurons (PN) developed higher selectivity compared to the formerly-inhibitory interneurons (Int) (Figure 7, orange), consistent with our previous finding that principal neurons have higher selectivity regardless of the sign of their outputs.

Again consistent with previous results, in the *NoConstraint* networks, the principal neurons no longer have sparser connectivity, instead, they are almost fully connected (Figures 7, 14). This result again indicates that the link between connectivity and selectivity depends crucially on whether Dale's law is applied. In a network without Dale's law, a neuron can receive inputs from all other neurons and remain selective, as long as the inputs from most neurons cancel out, leaving this neuron effectively driven by a small set of inputs.

**Impact of training and dataset**   The emergent differences between model E and I neurons is not only a result of the built-in E-I asymmetry, but also a result of the training process. Before training, model E and I neurons have the same selectivity and connectivity (Figure 5). During the training process, E neurons rapidly gained high selectivity, even before the network approaches high task accuracy (Figure 10). These results suggest that the emergence of high selectivity (and possibly sparse connectivity) is not necessarily dependent on the network performing object classification tasks well. To test this hypothesis, we trained networks to classify a set of randomly labeled CIFAR10 images (see Appendix K) (Zhang et al., 2016). After training, networks can achieve $20 - 50\%$ accuracy on the training set (chance level on the validation set, as expected). Higher selectivity and sparser connectivity among excitatory neurons still emerge in such networks (Figure 16), although less consistently across areas or random seeds. Understanding the minimal task necessary for the emergence of E-I differences would be interesting for future work.

## 4   DISCUSSION

We have shown that recurrent neural networks equipped with excitatory and inhibitory cells are capable of capturing several important features of the brain, including higher selectivity and sparser connectivity among excitatory neurons. These qualitative features emerged from the pressure to perform the task, suggesting that these qualities are indeed beneficial to task performance. This allows us to study what aspects of the network give rise to this distinction between excitatory and inhibitory neurons. We found that the higher selectivity of excitatory neurons is mainly driven by their role as the principal neurons that transmit information to upper layers. When Dale's law is obeyed, a higher selectivity necessitates sparser connectivity among excitatory neurons. Our

findings predict that if a brain area contains inhibitory principal neurons, these neurons should be more selective than interneurons of the same area.

Our model omits many elements of biological neural circuits, such as synaptic dynamics, spiking, long-range feedback connections, and so on. Our work aims at exploring the necessary computational conditions for several experimentally-observed properties to emerge. It is, in fact, another implication of our work that the biological features mentioned above are not required for such emergent properties. Rather than attempting to implement as many biologically-relevant network features in our models as we could, we include features that represent a step forward in biologically fidelity relative to most deep networks (e.g., excitatory and inhibitory neurons), but which are still feasible to implement on high-dimensional visual classification tasks. It is an interesting challenge for ongoing and future work to further incorporate biological components, including those mentioned above, into deep neural networks.

Optimization algorithms like stochastic gradient descent combined with large datasets are highly effective at tuning connection weights to perform tasks. However, it remains unlikely to observe a specific set of desired structural principles emerge naturally through training. We designed our standard networks to obey Dale's law. We have not explored whether Dale's law could emerge out of training. Such an emergence is theoretically possible, but has not been demonstrated, to the best of our knowledge. There are at least three possibilities: (1) the optimization algorithm is not strong enough to discover this solution because it remains a exponentially small part of the solution space $(1/2^N)$ of the entire solution space, $N$ being the number of connection weights). Similarly, it would be difficult for a ResNet structure (He et al., 2016) to develop from training a vanilla 100+ layer deep feedforward network. (2) The task we used is not appropriate for the emergence of Dale's law. It is conceivable that Dale's law is beneficial to some tasks that have yet been identified. (3) Finally, it is possible that Dale's law is a result of a compromise that the brain has to strike due to its biological nature, and is irrelevant to general computing machines. Understanding the nature of the computational benefit of Dale's law (if any) would be a major achievement in computational neuroscience, and may shed light on better designs of artificial neural networks.

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

## A  ADDITIONAL DETAILS OF THE MODELS

Our models consist of two feedforward layers and two recurrent layers for CIFAR10. Convolutions in the feedforward layers are regular convolutions. In the recurrent layers, we use depth-separable convolution, where a spatial convolution is applied on each channel separately and then followed by a point-wise convolution that mixes channels (Chollet, 2017). Depth-separable convolution is used because using regular convolution for recurrent connections leads to substantially lower (often around chance-level) and less stable accuracy on ImageNet in our networks. When training on CIFAR10, networks with regular convolution can still reach around 85% accuracy, but we observed a greater variability across networks with different random seeds. Kernel size is 3x3 for all convolutions. To measure the connection density, the 4-d convolution matrix of size [3, 3, channels_in, channels_out] is recovered from the spatial and point-wise convolution kernels, and then averaged over the first two spatial dimensions. This would give us a channel-to-channel weight matrix, with a total of channels_in×channels_out connection weights. The results are similar when we do not average across space and instead analyze the 3×3×channels_in×channels_out connection weights. Density is quantified as the proportion of connection weights that exceed a chosen threshold. We picked a threshold $\exp(-10)$ that separates the strong mode from the weaker modes in the distribution of all connection weights. The precise value of the threshold does not impact our qualitative results. Max pooling is applied on the output of each layer. Pooling stride is 1 for the first feedforward layer, and 2 for other layers.

For ImageNet, the L2 weight regularization coefficient is 0.0001. The regularization strengths are taken directly from the official Tensorflow ResNet models for CIFAR10 and ImageNet. We found that, without weight regularization, the connection probability of both excitatory and inhibitory connections become close to 1, far from experimentally-observed values. The initial learning rate is 1 and decays 10-fold at epochs 30, 60, 80 and 90, in all 100 epochs. Height and width of input images are resized and cropped to 128 pixel. The network consists of two convolutional feedforward layers of 64, and 128 channels each, followed by three recurrent layers. The first recurrent layer contains 256 PN, 64 IG, and 64 OG channels. For each type of neurons, the number of channels is the same in the second recurrent layer and doubled in the third recurrent layer. Batch-normalization is applied to the cell state $C_t$. The network is unfolded for 5 time steps, and the classification output is read-out from the final time step using a fully connected linear layer. For all conditions, we trained 5 networks with different random seeds. The network reaches approximately 55% test accuracy on ImageNet.

## B  BIOLOGICAL RELEVANCE OF RELU UNITS

Besides being the standard choices of activation function for modern deep convolutional neural networks (Glorot et al., 2011; Nair & Hinton, 2010; He et al., 2016), ReLU and its close variants (ELU, LeakyReLU, etc.) are widely used in deep neural network models for visual neuroscience (many of them being the same models used in computer vision) (Rajalingham et al., 2018; Nayebi et al., 2018; Lindsey et al., 2019)

Admittedly, the ReLU activation function is not an accurate depiction of a biological f-I curve, however, it does capture the important rectification of activity at low input values. ReLU does not saturate at high input values, while neurons obviously do ($> 200$ sp/s). However, most cortical neurons do not operate in this high-activity saturating regime. Many rate-based computational models of neural circuits use activation functions that do not saturate at high values, e.g., Wong & Wang (2006); Rubin et al. (2015).

## C  COMPUTING ORIENTATION SELECTIVITY

A static oriented grating is a two-dimensional sinusoidal wave $G(x, y)$ satisfying:

$$G(x, y) = \cos\{k[-(x - x_0)\sin\theta + (y - y_0)\cos\theta] + \phi\},$$

where $\theta$ is the orientation, $k$ is the spatial frequency, $\phi$ is the phase and $(x_0, y_0)$ is the center location. We generate a total of 432 gratings (Figure 8a) using 18 orientations ($10°$ apart), 6 spatial frequencies (1, 2, 3, 4, 6, 8) and 4 phases ($90°$ apart).

We present each grating image to a network, and each neuron's preferred orientation, spatial frequency and phase is chosen when the neuron has the maximal activity (averaged over time step). At its preferred spatial frequency and preferred phase, the global Orientation Selectivity Index (gOSI) is computed based on all orientations. Unlike gOSI, the orientation skewness is computed based on all 432 grating images.

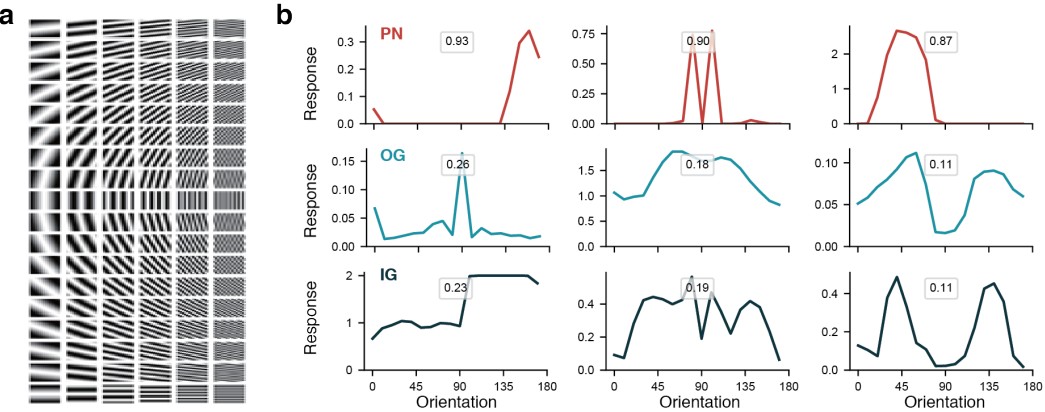

**Figure 8: Model excitatory neurons have higher orientation selectivity than inhibitory neurons.** (a) A total of 432 gratings, using 18 orientations (10° apart), 6 spatial frequencies (1, 2, 3, 4, 6, 8) and 4 phases (90° apart). (b) The orientation tuning curves of example PN, OG, and IG neurons from area 3 of one *Standard* network. The number in each panel indicates the neuron's global Orientation Selectivity Index (gOSI).

## D    MODEL VARIANTS

We tested a number of model variants to test the generality of our findings. The results are summarized (Figure 9) in the same style as Figure 4.

### D.1    READOUT AND GATING

In our *Standard* model, the logits are read out from the last time step of the final recurrent layer (area 4). We also trained networks where the logits are read out from the final recurrent layer's activity summed across all time steps. In Figure 9a, We denote the former structure as "LastT", and the latter structure as "SumT".

Apart from multiplicative (Mul) input gates and subtractive (Sub) output gates in the standard network, we implemented other combinations of gating mechanisms. In Figure 9a, a network is named as "Readout_InputGateOutputGate". For example, our *Standard* model is named "LastT_MulSub" in this plot.

### D.2    TRANSITIONING FROM *Standard* TO *StandardEI*

Figure 9b contains a set of models in transition from our *Standard* to *StandardEI* model. *NoOG_ECurrBN* removes the output gate of our *Standard* model and moves the batch normalization from cell state $C_t$ to the input current of PN neurons. Based on *NoOG_ECurrBN*, we obtained *NoOG_ECurrBN_SSub* by changing the multiplicative input gate into a simplified subtractive current. By furthering simplifying the forget gate structure to be the same as the *StandardEI* model, and adding recurrent structure for IG neurons, we get the *NoOG_ECurrBN_SSub_SFG_RecurrI* model, which is equivalent to the *StandardEI* model.

### D.3    BATCH NORMALIZATION AND DROPOUT

In Figure 9c, we tested various networks with or without batch normalization or dropout. In our *Standard* model, removing the batch normalization on cell states (*NoPNCellBN*) or replacing it with

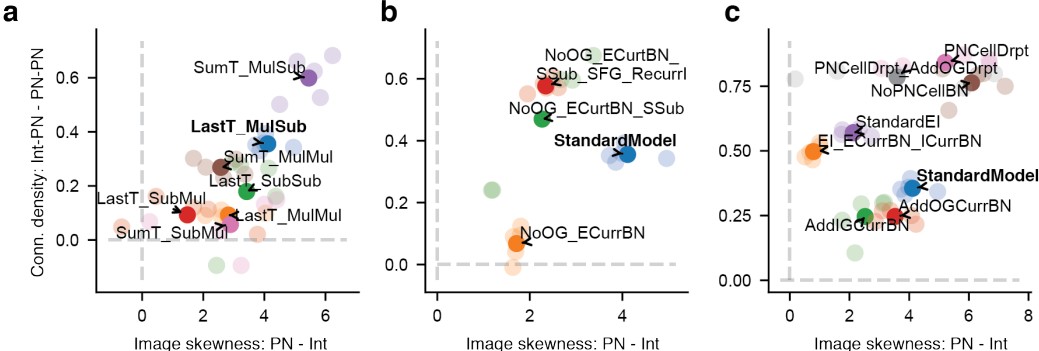

**Figure 9: The E-I differences emerge across a range of network variations.** (a-c) The E-I differences are summarized using the difference in image skewness between principal neurons (PN) and interneurons (Int), and the difference in connection density between interneuron-to-PN and PN-to-PN connections, in the same way as Figure 4. (a) Models with different readout and gating mechanisms. The *Standard* model is denoted here as the *LastT_MulSub* model. (b) Models in transition from the *Standard* model to the *StandardEI* model. The latter is denoted here as the *NoOG_ECurrBN_SSub_SFG_RecurrI* model. (c) Models with variations of batch normalization or dropout. All model variants develop the E-I differences. Results are combined from areas 3 and 4. Light circles: individual networks. Dark circles: average.

a dropout function (*PNCellDrpt*) (keep probability equals to 0.9) retains the performance above 80% and the E-I differences in both selectivity and density (Figure 9c).

For the *Standard* model with gating neurons, adding batch normalization for IG (*AddIGCurrBN*) or OG neurons (*AddOGCurrBN*) increases selectivity for the batch-normalized inhibitory neurons. Nevertheless, the inhibitory selectivity remains lower than that of PNs. The same trend is observed when batch normalization is added to inhibitory neurons in the *StandardEI* model (*EI_CurrBN_ICurrBN*), or when dropout is added to output-gating neurons in *PNCellDrpt* (*PN-CellDrpt_AddOGDrpt*).

## E    EXPERIMENTAL ESTIMATES OF E-I DIFFERENCES IN SELECTIVITY AND CONNECTIVITY

Here we intend to estimate the experimentally-observed differences in selectivity and connectivity across E and I neurons. Znamenskiy et al. (2018) recorded from mouse V1 responses to gratings. They quantified the response skewness of PV+ and PV- neurons (their Fig. 1b). PV, i.e. parvalbumin, is expressed only in a major subtype of inhibitory neurons. The difference between PV- and PV+ skewness is about 2.1. Song et al. (2005) measured the connection probability of pairs of excitatory neurons in rat V1. For pairs of cells with intersomatic distance lower than 100 microns, the connection probability is about 11% (their Fig. 2b). Packer & Yuste (2011) recorded from L2/3 of mouse barrel cortex (S1), and found that the connection probability from PV+ to pyramidal cells with intersomatic distance less than 200 microns is around 42% (their Fig. 4f). The last two results are obtained from different areas, species, and using different methodologies, so the quantitative value of the estimation is not to be trusted. However, the qualitative difference is robust.

## F    CLASSIFICATION PERFORMANCE DURING TRAINING

The classification performance of the *Standard* model on CIFAR10 increases rapidly in the first 10 epochs, and continues to improve after 150 epochs of training. The E-I difference in selectivity also increases rapidly in the first 10 epochs (Figure 5a,b), while the difference in connection density plateaus at about epoch 50 (Figure 5c).

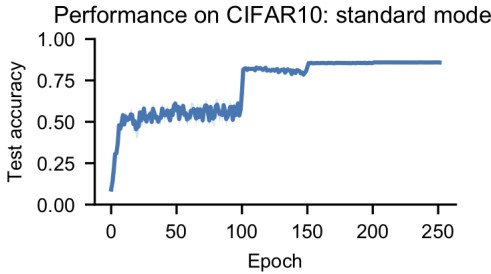

**Figure 10: The classification accuracy during training.** The learning rate is decayed at epoch 100, 150, and 200.

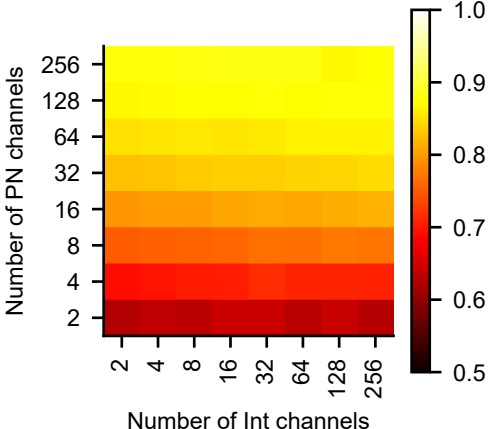

**Figure 11: Performance of networks with various numbers of excitatory (principal neuron, PN) and inhibitory (interneuron, Int) channels.**

## G  PERFORMANCE OF NETWORKS WITH VARIOUS NUMBERS OF EXCITATORY AND INHIBITORY CHANNELS

The performance of networks with various numbers of excitatory and inhibitory channels is summarized (Figure 11) in the same style as Figure 6. The performance depends mainly on the number of excitatory (PN) channels, rather than the number of inhibitory (Int) channels. All networks shown in Figure 11 and Figure 6 have accuracy above 60%.

## H  PERFORMANCE OF THE *InhPN* AND *NoConstraint* NETWORKS

The performance of the standard, *InhPN* and *NoConstraint* networks is summarized in Figure 12. Such *NoConstraint* networks achieve slightly better accuracy compared to the standard Dales law-obeying networks.

## I  DETAILED COMPARISON BETWEEN THE STANDARD MODEL AND ITS VARIANTS

The *Standard* model is compared to its *InhPN* variants (Figures 13) and the *NoConstraint* variants (Figure 14).

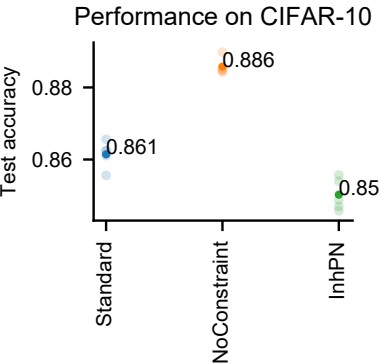

**Figure 12: Performance of the *InhPN* and *NoConstraint* networks compared with the standard models.**

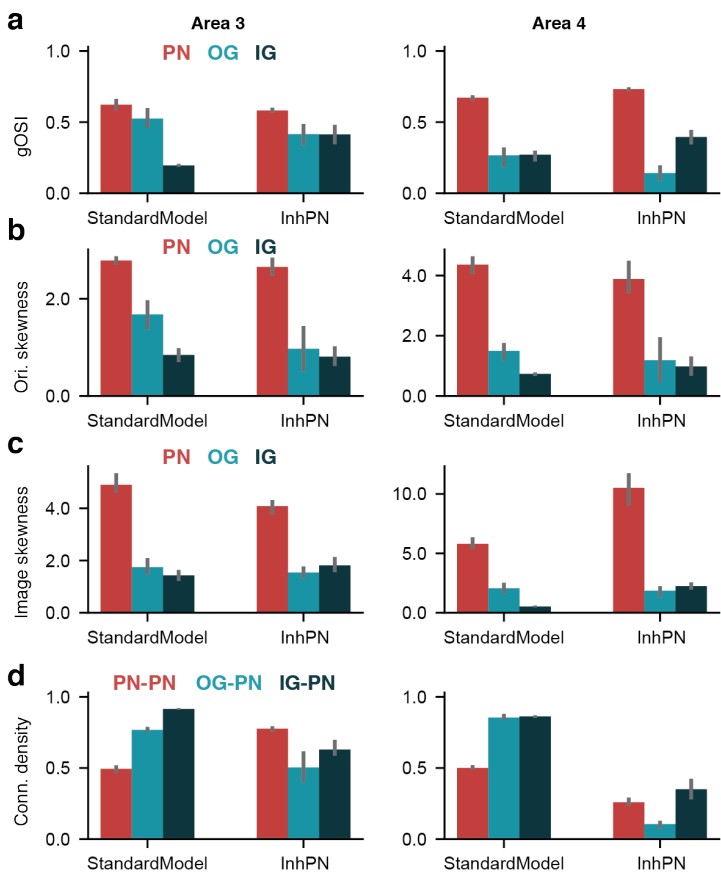

**Figure 13: Detailed comparison between the *Standard* model and *InhPN* network.** Comparing gOSI (a), orientation skewness (b), image skewness (c), and connection density (d) for areas 3 (left) and 4 (right) between the *Standard* model and the *InhPN* network.

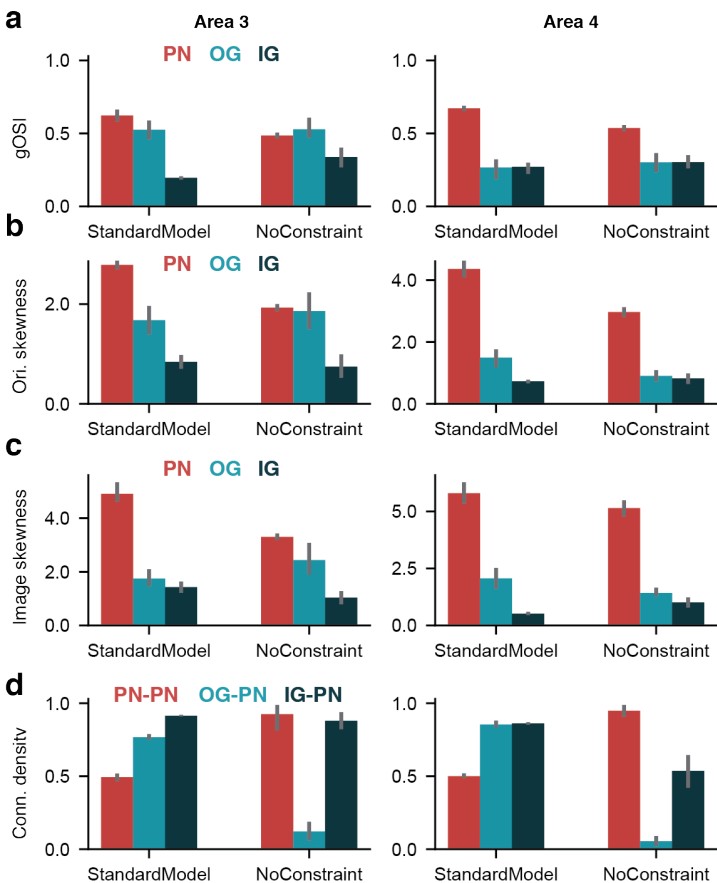

Figure 14: **Detailed comparison between *Standard* model and *NoConstraint* network.** Comparing gOSI (a), orientation skewness (b), image skewness (c), and connection density (d) for areas 3 (left) and 4 (right) between *Standard* model and the *NoConstraint* network. In the *NoConstraint* model, the OG-PN connectivity is close to zero, suggesting that OGs are only weakly participating in the computation.

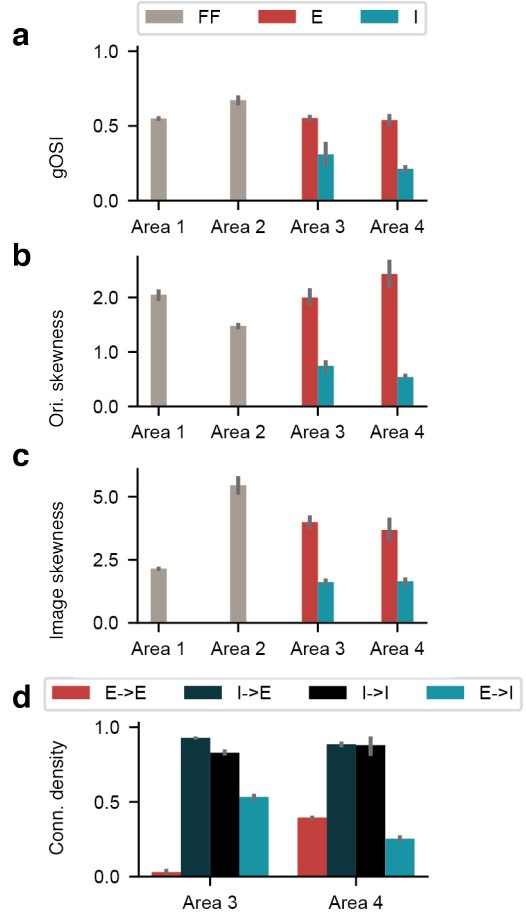

**Figure 15: Detailed analysis of the *StandardEI* model.**

## J    DETAILED ANALYSIS OF THE *StandardEI* MODEL

The *StandardEI* model shows similar qualitative trends (Figure 15) as the *Standard* model used in the main text (Figures 2, 3).

## K    TRAINING NETWORKS ON RANDOM LABELS

We trained networks on CIFAR-10 with shuffled labels (Zhang et al., 2016). Our *Standard* network for CIFAR10 contains $\sim 107K$ trainable parameters, compared to $\sim 1.6M$ parameters in networks used by Zhang et al. (2016), therefore we used shuffled training sets 10X smaller than the original training set. Unlike Zhang et al. (2016), networks trained to fit random labels did not have regularizations turned off. If we turn off L2 regularization on weights, then connection probability of excitatory and inhibitory neurons approach 1, similar to observed before (see Appendix A.)

## L    ANALYZING THE IMPACT OF REMOVING INHIBITORY NEURONS

In the *Standard* networks, the classification accuracy appears to depend very weakly on the number of inhibitory channels (Figure 11). To text if inhibitory neurons are needed at all, we trained the *Standard* and *StandardEI* networks with no inhibitory channels. To our surprise, the accuracy dropped, but rather moderately (Figure 17a) (Standard, from 0.855 to 0.817; StandardEI, from 0.842 to 0.728). We sought to understand what compensated for the loss of inhibitory neurons. We found

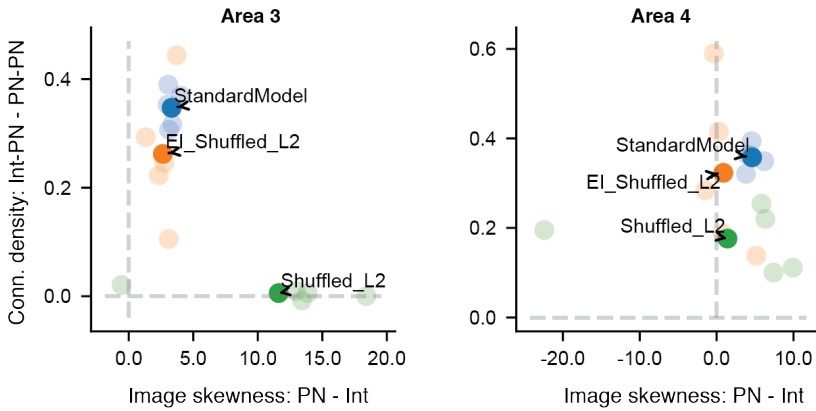

**Figure 16: Analysis of networks trained on shuffled labels.** L2 is used to emphasize that L2 weight regularization is intact in networks trained on shuffled labels, unlike Zhang et al. (2016). The *Shuffled_L2/EI_Shuffled_L2* networks achieve on average 43.9%/24.6% accuracy on the randomly-labeled training set.

that without Batch Normalization (BN) on excitatory neurons, the loss of inhibitory neurons becomes devastating (Figure 17b), reducing the performance to chance level (Standard, from 0.837 to 0.107; StandardEI, from 0.711 to 0.122). Therefore, inhibitory neurons are particularly necessary in networks trained without BN.

To ensure that the main findings are independent of whether batch normalization is applied, we repeated our experiments of Figs. 6 and 7, but for Standard and StandardEI architectures with no BN. We were able to reproduce all major findings (Figure 17d).

We did notice that even though having 0 inhibitory neurons is devastating, the accuracy can be rescued by having a single inhibitory channel (Figure 17c), which in a convolutional network corresponds to many inhibitory neurons with the same tuning. This is consistent with experimental findings that inhibitory neurons are weakly or non-selective.

We would also like to point out that, even with BN applied, inhibitory neurons are active and strongly connected with excitatory neurons, meaning that they participate in the proper functioning of the network.

To summarize, inhibitory neurons are always used, but only indispensable when BN is not applied. The main conclusions of the paper hold with or without BN.

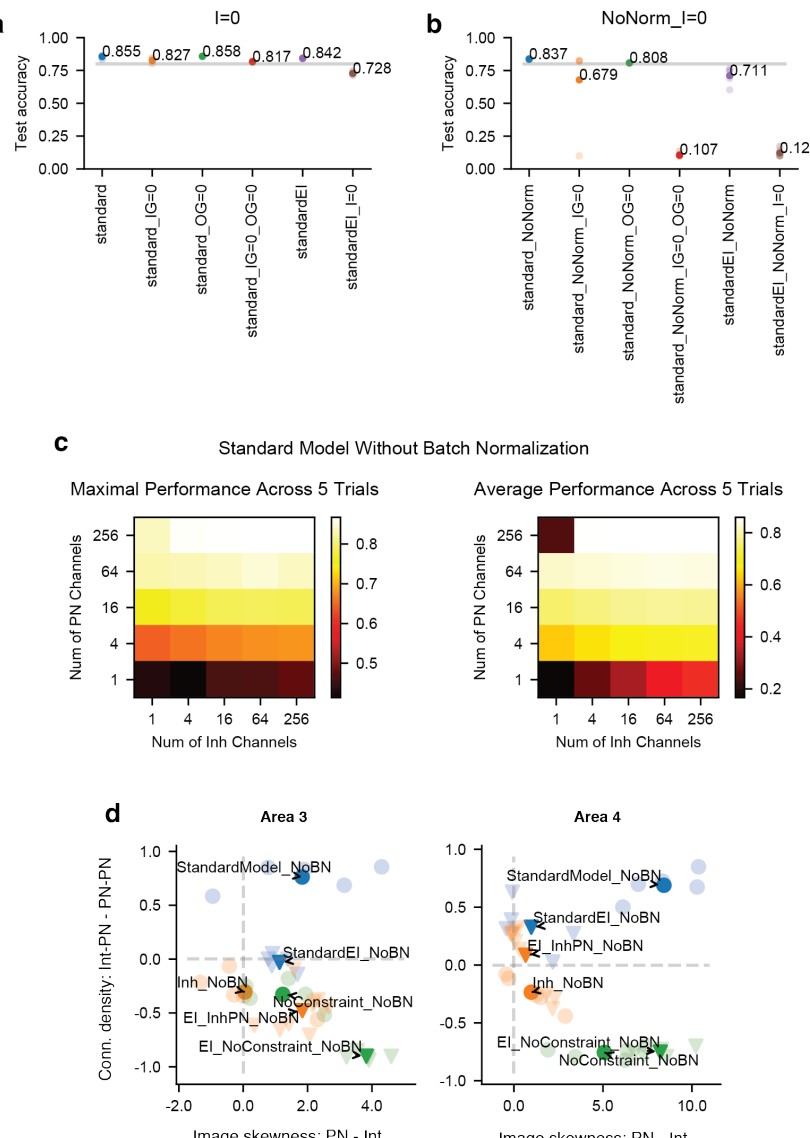

**Figure 17: Analyzing the impact of having no inhibitory neurons.** (a,b) Accuracy in networks without inhibitory neurons. (a) Standard and StandardEI networks, (b) Similar architectures, but without batch normalization (NoNorm). (c) Performance of networks with varying numbers of excitatory and inhibitory channels. Maximum (left) and average (right) performance across 5 random seeds. (d) *InhPN* and *NoConstraint* variants of networks without batch normalization reproduced findings from Figure 7.

