# OpenReview forum: "Understanding the functional and structural differences across excitatory and inhibitory neurons"
_ICLR.cc/2020/Conference — Reject_

### Official Review · AnonReviewer3 · 2019-10-23
**Official Blind Review #3**

**Rating:** 8

**Review:**

This paper explores why the brain might have separate excitatory (E) and
inhibitory (I) cells and how their different properties affect their
network connectivity.  The paper is novel and addresses an interesting
question.

The paper follows Dale's law and has separate populations of E and I
cells.  The cells differ in proportion (4 times more E cells) and in
projection patterns (E cells project to the next layer).  From these
constraints they found that some of the other observed differences
between E and I cells developed, specifically greater selectivity of E
cells and greater connectivity of I cells.

Both subtractive and divisive inhibition are modeled in the Standard model.

Through careful controls, they show that increased selectivity of E
neurons depends on having a lot of E neurons (but there was not a
corresponding effect for I neurons).  Likewise the sparse connectivity
of E neurons was due to a large number of E neurons.  They also showed
that being projection neurons was more important than being E cells
for strong selectivity.  Finally networks without Dale's law imposed
perform slightly better.

These are all interesting findings that will benefit the field of
neuroscience but may also lead to insights for the next architecture
boost to improve deep learning.  I am in favor of acceptance.





**Experience Assessment:**

I have published in this field for several years.

**Review Assessment: Checking Correctness Of Derivations And Theory:**

I assessed the sensibility of the derivations and theory.

**Review Assessment: Checking Correctness Of Experiments:**

I assessed the sensibility of the experiments.

**Review Assessment: Thoroughness In Paper Reading:**

I read the paper at least twice and used my best judgement in assessing the paper.

---

> ### Author Response · Authors · 2019-11-13
> **Response to Reviewer #3**
>
> We thank the reviewer for the feedback.
>
> Our work assesses what properties are necessary and sufficient for the emergence of sparse connectivity and high selectivity of excitatory neurons. It would be very important to understand why excitatory and inhibitory neurons segregate in the first place. Although this topic is not quite addressed in our work, it would be of tremendous interests for the future. We have clarified this point in the Discussion.

---

### Official Review · AnonReviewer1 · 2019-10-23
**Official Blind Review #1**

**Rating:** 6

**Review:**

[Update after rebuttal period]

I would like to thank the authors for the detailed response and for addressing the concerns on such a tight schedule.

Overall my two concerns appear to have been right: 1. The object classification task is not really relevant to elicit the observed behavior and 2. Inhibitory neurons are not essential (at least when training with batch norm).

Do the conclusions of the paper about understanding of E/I cells in the brain still hold? I am not really sure. I would like to urge the authors to carefully assess their conclusions again in light of the new evidence and perhaps rephrase abstract, intro and discussion where necessary.

Having said that, I still like the paper and its approach. I think it provides valuable insights to the field (e.g. neuro-inspired architecture design, concerns regarding training objectives, etc.) and important steps in the right direction and could be a valuable contribution even if the original claims are not entirely substantiated.


[Original review]

This paper seeks to understand why excitatory neurons in cortex are more stimulus selective than inhibitory cells and why excitatory cells are more sparsely connected to each other. The authors train a recurrent neural network model with a number of biological constraints on CIFAR-10. These constraints include Dale's law, an unequal number of E vs. I cells and only excitatory connections between layers (areas) of the network. I would summarize the main findings as follows:

- Excitatory principal neurons are more selective and more sparsely connected to each other than local inhibitory neurons, consistent with biology

- Stimulus selectivity and performance depend only on the number of excitatory cells, but neither on the E/I ratio nor the number of I cells

- High selectivity appears to be a general property of principal cells, but does not depend on the sign of local connections, while sparse connectivity is mainly linked to the excitatory local connections


Strengths:
+ Architectural decisions are well motivated from a neuroscientific perspective
+ Provides a hypothesis why certain connectivity and selectivity patterns emerge in the brain by incorporating biological knowledge into neural network models trained on a specific task
+ Well written and clear, logic development of the arguments

Weaknesses:
- Unclear whether classification task is necessary to elicit the authors' observations
- Interneurons seem unnecessary, raising concerns about relevance of results
- Also trains on ImageNet, but only some results are shown; most from CIFAR


Overall I like the paper from the perspective of a neuroscientist, as it provides a kind of normative account of why things in the brain might look the way they do. My enthusiasm is somewhat limited, however, because I am not convinced the classification task is actually what drives the authors' observations. I am willing to increase my score and argue more strongly for the paper if the authors can address this concern, detailed in the following:

A few observations lead me to believe the classification task the networks are trained may not be important to elicit the observed behavior.

1.) The emergent properties in stimulus selectivity show up almost immediately after training starts according to Fig. 5A+B. Performance, on the other hand, takes a few more epochs to pick up according to Fig. 10.

2.) The connectivity preferences that emerge in Figure 5C appear at already 50 epochs where CIFAR-10 performance is at 50%. To put that into perspective, a linear SVM already achieves 40% on this task, and a one-layer multilayer perceptron with only 100 hidden neurons reaches that performance.

3.) The results of Fig. 11 suggest that the number of inhibitory channels is not important. Did the authors try a stripped down version with only excitatory cells?

I suggest the authors train on CIFAR-10 with shuffled labels on the training set [Zhang et al. 2016; https://arxiv.org/abs/1611.03530]. Although performance on the test set would be at chance (no object recognition capabilities), it would be interesting to see whether the selectivity and connectivity properties still emerge.

Finally, (related to 3.) I wonder whether it makes sense to draw conclusions about differences between E and I cells from a model trained on a task where the number of inhibitory cells seems irrelevant. Wouldn't one need to have at least both types of neurons to be required for the task in order to draw such conclusions?


Minor Comments:

- I sometimes found the nomenclature of the multiple models you tried a bit confusing and hard to follow — especially in parts of the Appendix

- The type of operations (convolution, element-wise) in equation 1, together with the meaning of nonlinearities like \sigma_c are only defined one page after, making that section a bit hard to follow.

**Experience Assessment:**

I have published in this field for several years.

**Review Assessment: Checking Correctness Of Derivations And Theory:**

I assessed the sensibility of the derivations and theory.

**Review Assessment: Checking Correctness Of Experiments:**

I carefully checked the experiments.

**Review Assessment: Thoroughness In Paper Reading:**

I read the paper thoroughly.

---

> ### Author Response · Authors · 2019-11-13
> **Response to Reviewer #1 - On whether classification task is important and inhibitory neurons are necessary**
>
> We thank the reviewer for the very thoughtful observations and feedback. We have run multiple preliminary experiments that hopefully address the reviewer’s concerns. We included these preliminary results in two new supplementary figures (Figure 16, 17) and sections (Appendix J, K). Given the short amount of time available for rebuttal, these figures are far from publication quality yet. We will continue to improve them.
>
> 1. On whether the classification task is important.
> The reviewer made an astute observation about the earlier rise of orientation and image skewness compared to classification performance. To draw attention to this phenomenon, we have edited the manuscript text. This phenomenon can have at least two causes. One possibility is that the rise of skewness is unrelated to classification per se. Another possibility is that the rise of skewness is necessary for subsequent improvement in classification accuracy.
>
> To test these hypotheses, as suggested by the reviewer, we trained networks on CIFAR-10 with shuffled labels closely following the method of Zhang et al. 2016 (Appendix J, Figure 16). Our Standard network for CIFAR10 contains 107K trainable parameters, compared to ~1.6M parameters in networks used by Zhang et al. 2016, therefore we used shuffled training sets 10X smaller than the original training set. Unlike Zhang et al. 2016, networks trained to fit random labels did not have regularizations turned off. If we turn off L2 regularization on weights, then connection probability of excitatory and inhibitory neurons approach 1, similar to observed before (see Appendix A).
>
> After training, networks can achieve high accuracy on the training set (but not on the validation set, as expected). Higher selectivity and sparser connectivity among excitatory neurons still emerge in such networks (Figure 16), although less consistently across areas or random seeds. Understanding the minimal task necessary for the emergence of E-I differences would be interesting for future work. We have added a Results section to show this finding.
>
> 2. On whether inhibitory neurons are necessary.
> The reviewer made another excellent observation that the classification accuracy appears to depend very weakly on the number of inhibitory channels. As suggested by the reviewer, we trained the Standard and StandardEI networks with no inhibitory channels (Appendix K, Figure 17). To our surprise, the accuracy dropped, but rather moderately (Standard, 0.855 —> 0.817; StandardEI, 0.842 —> 0.728). We sought to understand what compensated for the loss of inhibitory neurons. We found that without Batch Normalization (BN) on excitatory neurons, the loss of inhibitory neurons becomes devastating, reducing the performance to chance level (Standard, 0.837 —> 0.107; StandardEI, 0.711 —> 0.122). Therefore, inhibitory neurons are particularly necessary in networks trained without BN.
>
> To ensure that the main findings are independent of whether batch normalization is applied, we repeated our experiments of Figs. 6 and 7, but for Standard and StandardEI architectures with no BN. We were able to reproduce all major findings summarized by the reviewer.
>
> We did notice that even though having 0 inhibitory neurons is devastating, the accuracy can be rescued by having a single inhibitory channel (Figure 17c), which in a convolutional network corresponds to many inhibitory neurons with the same tuning. This is consistent with experimental findings that inhibitory neurons are weakly or non-selective.
>
> We would also like to point out that, even with BN applied, inhibitory neurons are active and strongly connected with excitatory neurons, meaning that they participate in the proper functioning of the network.
>
> To summarize, inhibitory neurons are always used, but only indispensable when BN is not applied. The main conclusions of the paper hold with or without BN. These new results are included in the updated manuscript and Figures.
>
> 3. On the results from ImageNet
> We did not run the same numbers of experiments as we did for CIFAR10, but we will include a summary figure in the same style as Figure 7 for networks trained on ImageNet.
>
> 4. On confusing nomenclature and explanation of Eq. 1
> We will update the figures with hopefully clearer names, including the ones in the Appendix. We will also update the manuscript to clarify the section of Eq. 1.

---

> > ### Comment · AnonReviewer1 · 2019-11-15
> > **Quick question**
> >
> > Thank you for the clarifications. One quick question: in Fig. 16, has L2 normalization been turned on or off? Text + caption in the PDF say the networks have been regularized; your rebuttal text says they haven't.

---

> > > ### Author Response · Authors · 2019-11-15
> > > **Sorry for the confusion**
> > >
> > > Thank you for catching this inconsistency. In short, we changed the training protocols but forgot to update the rebuttal text. Here's more detail.
> > >
> > > In our initial attempt at training networks to fit random labels, we followed closely the protocols in Zhang et al 2016 by turning all regularizations off. This had resulted in networks where both E and I connection probability are close to 1, leading to an apparent difference from our Standard and StandardEI networks. These results were reported on Nov 12 in the initial response.
> > >
> > > However, even in the Standard and StandardEI networks, all connection probability approach 1 if no weight regularization is introduced (reported in Appendix A). To have a fair comparison between networks fitting original and random CIFAR10, we decided to add the same level of L2 weight regularization to the networks fitting random labels. Now we observe sparser E connectivity in these networks (Figure 16), although this difference in E and I connectivity is less robust than that observed in Standard networks. Further work remains to be done to truly understand the minimum task for the emergent E-I differences.
> > >
> > > We modified the reviewer response accordingly and updated a new manuscript on Nov 15, but forgot to change that line pointed out by the reviewer. Sorry for the confusion.

---

> ### Author Response · Authors · 2019-11-15
> **Minor edits to manuscript to include training accuracy on randomly-labeled dataset**
>
> We realized that we did not include training accuracy of networks trained on randomly-labeled dataset. We updated our manuscript (Results section and Figure 16) to include these details:
>
> The Shuffled_L2/EI_Shuffled_L2 networks achieve on average 43.9%/24.6% accuracy on the randomly-labeled training set.

---

### Official Review · AnonReviewer2 · 2019-10-24
**Official Blind Review #2**

**Rating:** 6

**Review:**

The idea behind the paper is quite interesting and enticing. The authors attempt to emulate real-life human brain neural network structure with artificial neural networks similar in architecture to the visual cortex.
The authors claim that they observe similar patterns emerge in the Neural Networks when it comes to the interaction and topological properties of excitatory and inhibitory neurons.

Unfortunately, the paper's claims seems to be undermined by several factors. First, the properties of the artificial neural networks that the authors build into the model (excitatory/inhibitory neurons relative abundance, inter-layer projection) are likely to be sufficient to explain the properties authors claim are informative in their model.

In addition to that, some architectural choices by the authors are highly disputable and counter-intuitive. For instance ReLU activation for Neurons would be contradicting any biological reality.

Besides, to be biologically innovative, the authors' approach seems to miss a number of crucial factors - notably interaction between neurotransmitters, timing of discharges and synapses offsets, allowing single neurons to perform complex logical computations requiring artificial neural networks with simple scalar non-linear activation functions multi-layer sub-networks to learn.

The case the authors cite - similarity of architectures of convolutional neural networks and the pinwheel architecture of visual cortex V1/V2 areas, is one of the few examples of convergent architectures of real-world and artificial neural networks specializing in a task.

Finally, the role of Inhibory neurons in the visual cortex seems to be
well understood, both biologically and mathematically (see for instance https://www.sciencedirect.com/science/article/pii/S0896627303003325 or https://www.sciencedirect.com/science/article/pii/S0896627303003325 or https://www.sciencedirect.com/science/article/pii/S092842570300072X)

Overall, the quality of the article at least in its current state, does not seem to be ready for acceptance to ICLR, but I'm willing to adjust my opinion after reading the opinion of more qualified reviewers' in this area and the authors response.

**Experience Assessment:**

I have read many papers in this area.

**Review Assessment: Checking Correctness Of Derivations And Theory:**

N/A

**Review Assessment: Checking Correctness Of Experiments:**

I assessed the sensibility of the experiments.

**Review Assessment: Thoroughness In Paper Reading:**

I read the paper at least twice and used my best judgement in assessing the paper.

---

> ### Author Response · Authors · 2019-11-13
> **Response to Reviewer #2, Part 1 - Motivation of work**
>
> We thank the reviewer for the feedback. We have edited the manuscript to address the reviewer’s concerns.
>
> 1. On the motivation of this work
> We thank the reviewer for observing that  “the properties...build into the model...are likely to be sufficient to explain the properties authors claim are informative in their model.”  This is indeed exactly what we are attempting to demonstrate. Our results show that the built-in properties (abundance of E neurons, excitatory long-range projections) are sufficient to explain the emergence of other biologically-realistic properties (sparser E-E connections, more selective E neurons) not present at initialization. Training is required for the built-in properties to foster the development of these emergent properties. This can be seen in Figure 5, where there is no difference between selectivity and connectivity of E and I neurons at the beginning of training.
>
> One of our goals is to find precisely what the reviewer references: the minimal set of network characteristics that are required to produce biologically-realistic connectivity and selectivity through training. This is why we systematically trained different variants of the original model (Fig. 6, 7). We edited our Introduction section to clarify these points.

---

> > ### Author Response · Authors · 2019-11-13
> > **Response to Reviewer #2, Part 2 - Architectural choices**
> >
> > 2. On architectural choices
> > We thank reviewer #2 for noting the tradeoff between biological fidelity and network performance. We believe our architectural choices are reasonable (for example, reviewer #1 notes that  “Architectural decisions are well motivated from a neuroscientific perspective”), and are close to that of other papers recently published at ICLR and similar conferences. We detail our reasoning below. We have edited our Results section, and included a new Appendix section to discuss the architectural choices.
> >
> > 2.1 ReLU activation function
> > Besides being one of the standard choices of activation function for deep convolutional neural networks since around 2011 [Glorot, Bordes, Bengio 2011, Nair & Hinton 2010, He, Zhang, Ren, Sun 2016], ReLU and its close variants (ELU, LeakyReLU, etc.) are widely used in deep neural network models for visual neuroscience (many of them being the same models used in computer vision) [Rajalingham, Issa, DiCarlo et al. 2018 JNeuroSci, Cadieu, DiCarlo et al. 2014 PlosComBio, Sinz, Tolias et al. 2018 NeurIPS, Nayebi, Bear, Yamins, et al. 2018 NeurIPS, Lindsey, Ocko, Ganguli, Deny 2019 ICLR].
> >
> > Admittedly, the ReLU activation function is not an accurate depiction of a biological f-I curve, however, it does capture the important rectification of activity at low input values. ReLU does not saturate at high input values, while neurons obviously do (>200 sp/s). However, most cortical neurons do not operate in this high-activity saturating regime. Many rate-based computational models of neural circuits use activation functions that do not saturate at high values [e.g., Wong & Wang 2006 JNeuroSci, Rubin, Van Hooser, Miller 2015 Neuron].
> >
> > 2.2 More complex single neuron and synapse model
> > The reviewer cites a number of biological factors not included in our model, including “interaction between neurotransmitters”, “timing of discharges”, and single neurons that “perform complex logical computations”.
> >
> > As mentioned above, our work aims at exploring the necessary computational conditions for several experimentally-observed properties to emerge. It is, in fact, another important implication of our work that the biological features mentioned above are not required for such emergent properties. Rather than attempting to implement all biologically-relevant network features in our models, we include features that represent a step forward in biologically fidelity relative to most deep networks (e.g., excitatory and inhibitory neurons), but which are still feasible to implement on high-dimensional visual classification tasks. It is an interesting challenge for future work to further incorporate biological components, including those mentioned above, into deep neural networks. We have included this discussion in the Discussion section.
> >
> >
> > 3. On Pinwheel and the role of inhibitory neurons
> > “The case the authors cite - similarity of architectures of convolutional neural networks and the pinwheel architecture of visual cortex V1/V2 areas”. However, we would like to note that we did not mention pinwheels in our manuscript. In fact, convolutional neural networks can not develop pinwheel structures because of the convolutions, meaning that neurons selective to one direction would be located at all spatial locations.
> >
> > The reviewer mentioned that “The role of Inhibitory neurons in the visual cortex seems to be
> > well understood”, and cites two papers published in 2003 (2 of the 3 references cited by the reviewer are the same). We respectfully disagree with this assessment. The continued progress in our understanding of the role of inhibitory neurons, especially different types of inhibitory neurons, is evidenced by many high-profile publications studying them in the visual cortex published long after 2003 [e.g., Adesnik, Scanziani, et al. 2012 Nature, Atallah, Scanziani et al. 2012 Neuron, Fu, Stryker et al. 2014 Cell]. Nevertheless, in the Introduction, we now better acknowledge the classical literature on the functional role of inhibitory neurons by including more references from authors like David McLaughlin and Bob Shapley.

---

> > > ### Comment · AnonReviewer1 · 2019-11-15
> > > **I agree with the authors**
> > >
> > > I agree with the authors that their choice of architecture etc. makes sense. I don't think the suggestions to include more low-level details or biological realism would lead to additional insights. For the questions addressed in the paper, the level of detail is sufficient.

---

> ### Comment · AnonReviewer2 · 2019-11-15
> **Thanks for the didactic clarification (review upgraded)  Restore**
>
> As a reviewer who is less familiar with this area, I would like to thank you for the didactic clarification and thanks to the other reviewers for their useful feedbacks. My review has been upgraded accordingly.

---

> > ### Author Response · Authors · 2019-11-15
> > **Thank you for helping us improve the manuscript**
> >
> > We would like to thank the reviewer again for helping us make our manuscript more readable for a broader audience. The reviewer is right to point out that many common architectural choices in deep learning may appear or be unjustified from a biological perspective, and more efforts are warranted to discuss the reasoning behind and the consequences of such architectural choices.

---

### Author Response · Authors · 2019-11-15
**Update to the manuscript**

We would like to thank all reviewers for taking the time to review our work and providing insightful feedback.

We have updated our manuscript with edits throughout the main text and appendix, with two new figures and three new sections in the Appendix. The new materials are highlighted in blue for your convenience. The responses to reviewers are updated accordingly.

Several promised changes to the manuscript have yet to be made. We will continue to work on them. Thank you again.

---

### Decision · Program_Chairs · 2019-12-19

**Decision:**

Reject

**Comment:**

This paper explores the role of excitatory and inhibitory neurons, and how their properties might differ based on simulations.  A few issues were raised during the review period, and I commend the authors for stepping up to address these comments and run additional experiments.  It seems, though, that the reviewer's worries were born out in the results of the additional experiments: "1. The object classification task is not really relevant to elicit the observed behavior and 2. Inhibitory neurons are not essential (at least when training with batch norm)."  I hope the authors can make improvements in light of these observations, and discuss their implications in a future version of this paper.